# Metabolic multireactor: Practical considerations for using simple oxygen sensing optodes for high-throughput batch reactor metabolism experiments

**Matthew H. Kaufman** [1☯]*, **Joshua Torgeson**[2☯], **James C. Stegen**[1]

**1** Pacific Northwest National Laboratory, Earth and Biological Sciences Division, Richland, WA, United States of America, **2** Pacific Northwest National Laboratory, Energy and Environment Division, Richland, WA, United States of America

☯ These authors contributed equally to this work.
* matthew.kaufman@pnnl.gov

**Data Availability Statement:** The data collected for this manuscript is available from the ESS-DIVE

## Abstract

We present a system for carrying out small batch reactor oxygen consumption experiments on water and sediment samples for environmental questions. In general, it provides several advantages that can help researchers achieve impactful experiments at relatively low costs and high data quality. In particular, it allows for multiple reactors to be operated and their oxygen concentrations to be measured simultaneously, providing high throughput and high time-resolution data, which can be advantageous. Most existing literature on similar small batch-reactor metabolic studies is limited to either only a few samples, or only a few time points per sample, which can restrict the ability for researchers to learn from their experiments. The oxygen sensing system is based very directly on the work of Larsen, et al. [2011], and similar oxygen sensing technology is widely used in the literature. As such we do not delve deeply into the specifics of the fluorescent dye sensing mechanism. Instead, we focus on practical considerations. We describe the construction and operation of the calibration and experimental systems, and answer many of the questions likely to come up when other researchers choose to build and operate a similar system themselves (questions we ourselves had when we first built the system). In this way, we hope to provide an approachable and easy to use research article that can help other researchers construct and operate a similar system that can be tailored to ask their own research questions, with a minimum of confusion and missteps along the way.

## 1. Introduction

Metabolism of organic material in both fresh and saline systems is a critical yet poorly understood contributor to global carbon and nutrient cycling and greenhouse gas production. Carbon dioxide production varies significantly both spatially and temporally in tidal and freshwater systems with inner estuarine waters, tidal flats, and saltwater marsh sediments

Data Archive (https://data.ess-dive.lbl.gov/datasets/doi:10.15485/1985920).

**Funding:** A portion of the research described in this paper was conducted under the Laboratory Directed Research and Development Program at Pacific Northwest National Laboratory, a multi-program national laboratory operated by Battelle for the U.S. Department of Energy. MK was grateful for the support of the Linus Pauling Distinguished Postdoctoral Fellowship program. This research was supported by the U.S. Department of Energy, Office of Science, Office of Biological and Environmental Research, Environmental System Science (ESS) Program. This contribution originates from the River Corridor Scientific Focus Area (SFA) project at Pacific Northwest National Laboratory (PNNL). This research was supported under award DESC0018042. The sponsors and funders did not play any role in the study design, data collection and analysis, decision to publish, or preparation of the manuscript.

**Competing interests:** The authors have declared that no competing interests exist.

producing 17 to 17,000 tons $CO_2 \cdot km^{-2} \cdot yr^{-1}$ globally and freshwater littoral sediments producing up to 25% of global carbon dioxide [1, 2]. Freshwater organic carbon (OC) consumption, and therefore freshwater $CO_2$ production, is predominantly controlled by hydrobiogeochemical changes in aquatic systems [3].

However, understanding the role of metabolism in these large, complex environmental systems is logistically challenging due to the combination of complex field methods and natural heterogeneity in aquatic systems [4]. Field metabolism measurements have been carried out for decades in myriad locations. Such studies are often approached either through open-water whole-system approaches based on oxygen time series data [5–11] and many others, or by attempting to isolate the components of the system (hyporheic, benthic, planktonic, etc.) separately [12–17] and many others. Whole-system approaches interrogate processes across relatively large scales (meters to hundreds of meters) which integrates the influence of small-scale heterogeneity; however, such whole-system measurements provide a limited understanding of the individual components contained within the larger system. The scale and complexity of the systems interrogated by whole system approaches also make manipulative experiments difficult or impossible. Isolation of components in the field allows a more thorough understanding of the relative contributions of each component; however, the heterogeneity of natural systems, particularly sediment systems, drives a need for collection of field measurements at an impractically large number of locations and, like whole-system approaches, manipulative experiments (that is, experiments where a system is intentionally subjected to a specific set of conditions, rather than an experiment where field conditions are left largely uncontrolled) are logistically challenging.

To better understand the response of individual metabolic components of aquatic systems to changing environmental conditions such as temperature, sea level rise, or organic matter supply associated with global change, manipulative experiments are required. A great deal of information can be gleaned from small bioreactor experiments using sediment and water collected from the field and manipulated and measured in the lab [18]. However, these experiments are hampered by the high effort, low throughput, and low temporal resolution of existing techniques.

One of the most important pieces of information gathered from these small bioreactors is the rate of metabolism. Aerobic metabolism rates are frequently reflected in oxygen consumption rates, although inorganic carbon remineralization and abiotic reoxidation of reduced species can also contribute to oxygen consumption rates. There are several ways that dissolved oxygen (DO) can be measured in small bioreactors, but even the best existing methods are often expensive, time consuming, and provide limited time resolution. Most laboratory sediment oxygen consumption studies use some variation of a sealed reactor with intrusive oxygen concentration measurements at a small number of time points [19–21], or some form of flow-through reactor [22]. Some high-throughput methods for $CO_2$-based metabolic assays exist, however they do not allow for oxygen measurement [23].

There are a number of ways to measure DO concentrations, and optical DO sensors (optodes) are used widely in aquatic research as well as terrestrial and marine systems, both in field and lab settings [24–26]. Oxygen optodes have been used in many lab-scale experimental systems, however they are traditionally used primarily to obtain two-dimensional data in flumes [27–29] or tanks [30–35], or 1- or 0- dimensional (point) data in columns [36], or other advective-transport-driven applications [37], or in the form of microsensors [38–40]. In this manuscript we describe the application of an existing ratiometric oxygen optode technology [41] to a set of custom, flexible, and simple-to-produce small bioreactors. The optodes are mounted in the lids of the bioreactors, which are continuously rotated, providing 0-dimensional measurements of the oxygen concentration inside the well-mixed bioreactor over time. In particular, we detail the practical aspects involved with the construction and operation of

the "metabolic multireactor", a system for carrying out small bioreactor oxygen consumption experiments. The highly parallel system design, as well as the use of predominantly off-the-shelf parts, can provide users the opportunity to carry out large numbers of manipulative reactor experiments with reduced costs and increased throughput compared to, or in concert with, other available methods.

## 2. Methods

### 2.1 Method summary

The purpose of this metabolic multireactor system is to provide a relatively simple platform for batch reactor metabolism studies, and in particular for monitoring changes in DO concentrations over the course of experiments ranging from minutes to weeks or longer, with high temporal resolution. The system is designed for both high throughput of parallel bioreactor incubations as well as flexibility of bioreactor construction, allowing the system to be tailored, through the addition of other sensors and modification of the reactors themselves, to specific experimental needs and analyses in addition to oxygen consumption.

The core of the multireactor system is an expandable array of small tubular bioreactors with built-in oxygen optodes housed on a bed of rollers and imaged all at once with a modified DSLR camera (Fig 1). The system is highly scalable, as the roller units can be stacked in racks that put up to several hundred tubes within the field of view of the camera. The system is also highly flexible, as it can be configured with different types of tubular bioreactors and additional sensors, depending on the other analyses that might be of interest. The core oxygen optode technology used in the system is well-studied, and we do not intend to delve deeply into that aspect of the system. Instead, the aspects presented in this study consist of practical items that users are likely to find useful or have questions about when they design and construct similar systems, modified to explore their specific scientific investigations.

### 2.2 Optode production and application

The oxygen optodes applied to the bioreactors are constructed using a well-established method presented by Larsen, Borisov [41]. They consist of a cocktail of sensing dye and antenna/reference dyes as described by Larsen, Borisov [41], however platinum-tetrakis-pentafluorphenyl-porphyrin (PtTFPP) replaces the Platinum-octaethyl-porphyrin (PtOEP) that Larsen, Borisov [41] use as the sensing dye [42]. Macrolex Yellow GN is used in a dual role as both antenna and reference dye. Additional dye cocktail details are available in [41]. The dye cocktail is then processed onto the appropriate substrate for the type of bioreactor being produced. If necessary, a second layer of silicone and carbon is applied over the top of the sensor cocktail to prevent reflections from angular sediment grains and to reduce the potential confounding effect of fluorescence of naturally present organic molecules, again as described by Larsen, Borisov [41]. Applying the optode chemistry directly to acrylic sheets can reduce sensitivity [43] and introduce calibration drift [44] under some circumstances. These issues did not reduce the utility of our system for the described purpose, however Koren, Borisov [43] and Badocco, Mondin [44] provide additional methods to improve performance if necessary. The response of the system to varying oxygen concentrations is described by a modified Stern-Volmer equation presented by Larsen, Borisov [41] and Klimant, Meyer [24], and presented here:

$$\frac{R}{R_o} = \left[ \alpha + (1 - \alpha)\left(\frac{1}{1 + K_{SV} \cdot C}\right) \right] \qquad \text{Eq 1}$$

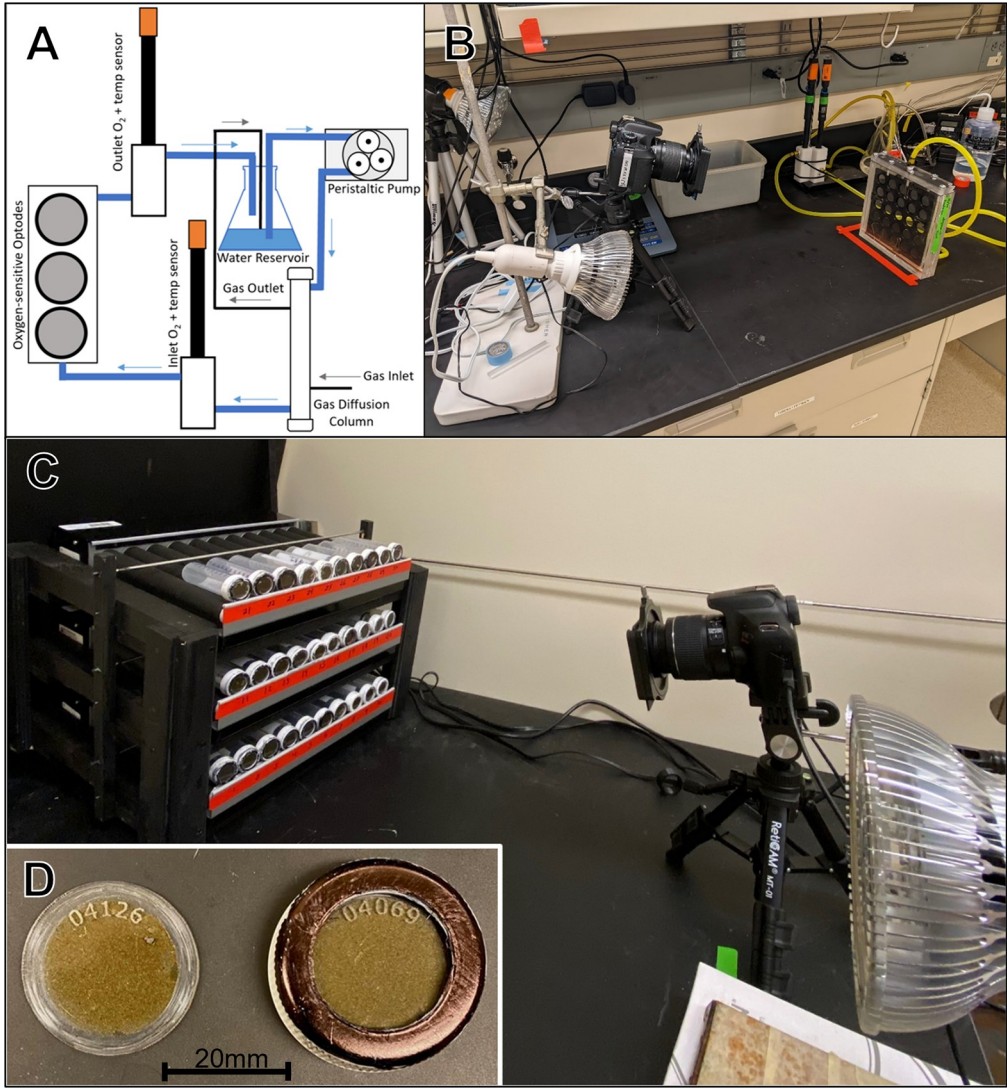

**Fig 1.** (A) Flow cell testing and calibration system used for controlling oxygen concentrations. Blue lines are water flow and black lines are gas flow. Gas is made up of nitrogen and air mixed by mass flow controllers. (B) Imaging setup. Camera and light are on the left, while multi-disk calibration cell is on the right. (C) Multireactor system in use. Note racks of reactor tubes on rollers at left, camera and excitation light at right. (D) detail of 28.2mm 50ml centrifuge-cap optode disk alone (left) and mounted in tube-cap (right).

Where $R$ is the (red-green)/green luminance ratio, $R_0$ is the ratio absent oxygen, $C$ is the oxygen concentration, $\alpha$ is the nonquenchable fraction of the signal, and $K_{SV}$ is the Stern-Volmer quenching constant. The use of Larsen, Borisov [41]'s ratiometric measurement approach is critical to the success of this system. Because the measured value is a function of both the red emission light (oxygen sensitive) and green emission light (oxygen insensitive), the system is unaffected by small variations in the excitation light intensity and dye coating thickness.

Tube-cap optodes (Fig 1) were prepared from 15.24 cm x 30.48 cm (6 in x 12 in) panels of 1.6mm (1/16 inch) clear scratch- and UV-resistant cast acrylic (Poly(methyl methacrylate), PMMA) sheet (part number 8560K178, McMaster-Carr, Elmhurst IL, USA). The acrylic sheet was spray-coated first with the dye cocktail, and then 24 hours later with the silicone and carbon cocktail. Spray coatings are prepared by diluting the cocktails 4:1 (in toluene for the dye

cocktail and in hexane for the silicone and carbon cocktail). The diluted coatings were sprayed onto the acrylic sheet using a central pneumatic 120 $cm^3$ high volume low pressure touch up air spray gun (part number 62300, Harbor Freight Tools, Calabasas CA, USA) operated at approximately 1.38 bar (20psi). After curing for 24 hours, the sheet was cut into disks using a 40 watt $CO_2$ laser cutter (part number HL40-5G-110, Full Spectrum Laser, Las Vegas NV, USA). Disks for 50 ml RNase-free plastic tubes (part number 28–106, Genesee Scientific, San Diego Ca, USA) were cut to 28.8mm dia, while disks for 40ml glass VOA vials (part number 12-100-107, Fisher Scientific, Waltham MA, USA) were cut to 21.15mm diameter. 21.15mm disks were also used for double-ended 40ml glass VOA vials, which were custom made by Precision Glassblowing (Centennial, CO, USA) from type 1 33-expansion glass VOA vials (part number 27184, MilliporeSigma, Burlington MA, USA). Prior to cutting the disks out of the sheet, the silicone and carbon layer was removed from a 4mm wide annular region of the surface using the laser cutter in raster mode. This step allowed the O-ring seal to bear against the dye coating, rather than the silicone-and graphite coating. If the O-ring bore against the silicone and carbon layer, there was a tendency for the O-ring to stick to the silicone layer and cause it to peel off of the dye layer, rendering the disk unfit for later recalibration and reuse without first re-applying the silicone layer. The laser cutter was also used to engrave a small identification number into each disk, allowing easy tracking and assigning individual calibration curves to individual disks. Tube-cap optodes were also prepared by cutting the disks out of the acrylic sheet first and then spraying them with the cocktails, using a 3d-printed mask to create the un-coated annular region, however this was much more laborious and time consuming than spraying the sheet prior to laser cutting and we recommend the coat-first technique.

Depending on the tube type, the standard tube cap needed to be modified to work with the tube-cap optode. For plastic vials, a 22.15mm (7/8 inch) hole was drilled in the cap using a hole saw (part number 4146a56, McMaster-Carr, Elmhurst IL, USA), and the internal sealing ridge was removed using a 28.58mm (1-1/8 inch) hole saw (part number 4146A61, McMaster-Carr, Elmhurst IL, USA). The optode disk was placed into the cap, coated side facing the tube, and a 30.16mm (1-1/8 inch) OD silicone O-ring (part number 1182n12, McMaster-Carr, Elmhurst IL, USA) was placed between the optode disk and the tube. For glass vials, the existing septum was simply removed from the cap and replaced by the optode disk and a 17.46mm (11/16 inch) OD O-ring (part number 1182N115, McMaster-Carr, Elmhurst IL, USA).

## 2.3 Calibration and test system

To test and calibrate the optode disks, a system for providing water with a known and controllable concentration of DO was necessary. To control the oxygen concentration of water in contact with the reactive surface of the tube-cap optodes, water was pumped through a 222mm long by 43mm dia. 3M™ Liqui-cel™ MM Series Membrane Contactor (part number 70201351114, 3M, St. Paul MN, USA). Two 0–1 LPM digital mass flow controllers (MassTrak 810, Sierra Instruments, Monterey CA, USA) were used to control nitrogen and air flow rates. Nitrogen and air leaving the mass flow controllers were mixed and fed through the membrane contactor. Waste gas from the membrane contactor was discharged into the headspace of the reservoir/bubble trap. A peristaltic pump (Kamoer UIP, Kamoer Fluid Tech, Shanghai, China) was used to control water flow rates through the membrane contactor and through the flow cell system. The pump pulled water from a 1L flask reservoir and passed it through the membrane contactor, through a flow cell containing a commercial DO sensor (part number 4410, YSI Inc, Yellow Springs OH, USA), through the optode disk flow cell, through a second commercial DO sensor, and back into the 1L reservoir flask. The open-topped reservoir flask

provided a mechanism to allow gas bubbles to escape from the fluid loop (bubble trap), and provide makeup fluid volume to the system. The two commercial DO sensors were used to monitor DO concentrations before and after the optode disk flow cell to verify steady-state conditions, and to approximate oxygen concentrations, though the flow controllers were considered more accurate than the commercial DO sensors. The calibration flow cell system is depicted in Fig 1. A variety of calibration and testing flow cells were constructed, using 3d printed and solvent-welded acrylic parts. 3D printer files for various calibration parts can be found in the S1–S16 Files. These may be helpful for users constructing their own calibration systems.

## 2.4 Optode imaging system

The imaging setup (Fig 1) consisted of a blue LED (450-460nm) excitation light (part number B01M0CKUO7, Amazon, Seattle WA, USA) approximately 79cm from the base of the flow cell and 46cm above the surface of the bench, though these distances are not critical and should be adjusted to provide even lighting. A digital camera (EOS Rebel T6, Canon Inc, Tokyo, Japan) with an 18.0 megapixel sensor was used to capture images of the optodes. The internal infrared blocking filter was removed from this camera (filter removal performed by Life Pixel Infrared, Mukilteo WA, USA), and it was equipped with a 12.5mm square 530nm long pass filter (part number 54–654, Edmund Optics, Barrington NJ, USA) mounted in a filter holder (part number B01JLNU7EU, Amazon, Seattle WA, USA) equipped with a 3D-printed adapter. Camera control and image capture was handled using Look@RGB software and a trigger box (LED trigger light, https://imaging.fish-n-chips.de/). Primary data processing was carried out using ImageJ software, and consisted of identifying the sensor of interest in an image, drawing a circle over it, and extracting the average red and green light intensities for that region [41, 45]. While the exact resolution of the images depends on the camera, lens, and distance from the sensor disks, in general we achieved a resolution of better than 100 pixels per square millimeter.

## 2.5 Optode performance testing

28.8mm tube-cap optode disks were tested to determine general performance characteristics (such as limit of detection), as well as how performance and consistency was impacted by a series of variations in external conditions (including excitation light angle, autoclave sterilization, imaging camera, and others). For most of these tests the 28.8mm disks were inserted into vial tube caps, which were screwed on to a 3D-printed flow cell which was connected to the calibration system. Deionized water was pumped through the cell with $O_2$ saturation varied from 0 to 100% (at 20% intervals) to generate a calibration curve (referred to here as the response curve) which could be compared with reference calibration curves. Tests were performed with laboratory lighting off, unless specified otherwise. The aperture value (Av) and time value (Tv) of the camera used for imaging was adjusted to enable near sensor saturation under low oxygen concentrations (that is, the maximum red-channel brightness the camera was able to capture) to ensure a high signal to noise ratio and to make use of as much of the camera's brightness resolution as possible. Our typical AV was F/5.6, and TV was 0.4 seconds. Where it is used, the parameter R was calculated according to Eq 1 presented in Larsen, Borisov [41]. In situations where calibration results between different treatments appeared substantially similar, their similarity was assessed statistically using a Kolmogorov-Smirnov test (ks.test function in R [46]). If the Kolmogorov-Smirnov P value was less than 0.05 then we reject the null hypothesis that the two sets of samples come from the same distributions, indicating that the two results are significantly different.

**2.5.1 Ambient lighting testing.** In all other multireactor experiments presented here, the system was operated in a darkroom to ensure the incident light was generated only by the 450-460nm LED light. All other lights were turned off or covered to ensure consistent data. To test the influence of ambient indoor lighting on multireactor performance, calibration curves were collected with ambient room lights (common fluorescent tube ceiling lights) turned on and compared to calibration curves generated under darkroom conditions. Under both lighting conditions, the multireactor was set up in the same configuration, with LED excitation lighting to illuminate the oxygen optodes. To show differences between data processed using these calibration curves, theoretical R values were generated at regular intervals between 0.4 and 1.5 and these values were processed using each calibration curve to calculate DO concentrations.

**2.5.2 Excitation light angle testing.** To determine if the angle of excitation light influenced the response of the optode, the light was set at 0° (directly facing the optodes), 30°, and 45° angle to the optode disks. A response curve was collected at each angle and the response curves were compared.

**2.5.3 Sterilization: Autoclave and high pH testing.** To test how the disks stood up to autoclave cycles, two sets of three disks were autoclaved using 2 autoclave settings. The first set of three disks were sterilized inside an autoclave bag on a 60-minute gravity cycle; the second set of three was sterilized in 200mL of deionized water on a 30-minute liquid cycle. Response curves were collected before and after autoclaving for comparison.

In order to determine how exposure to high-pH solutions impacted future optode performance, after initial calibration, three optode disks (a, b, and c) were transferred to a flask filled with a pH 10.0 solution. This solution was prepared by adding 1mM sodium hydroxide to a 36mM solution of sodium sulfite until the desired pH was reached. Throughout the experiment, the pH was verified weekly. The flask was stored in the dark between measurements.

**2.5.4 Diffusion testing.** Diffusion rates through various vial and bottle types were tested by filling vials/bottles with deoxygenated water which was generated by sparging with nitrogen, capping the vials, and imaging every ~15 minutes for ~24 hours. Each vial type was tested in triplicate. Tested vial types included the plastic centrifuge tubes, glass voa vials, and double-ended glass voa vials with a septum cap on one end and an optode disk cap on the other. The vials were tested both static and on rollers to facilitate mixing.

Assuming sterile water starting at a low oxygen concentration inside the reactor, the evolving partial pressure of oxygen inside the reactor can be described with the following ordinary differential equation:

$$r \cdot (p_a - p_i) \cdot S = dp_i/dt \cdot H \cdot V \qquad \text{Eq 2}$$

The analytical solution to the above differential equation, recast in terms of concentrations (mg/l) rather than partial pressures (bar) is:

$$p_i(t) \cdot H = p_a \cdot H - (p_a - p_0) \cdot H \cdot \exp(-rHVt) \qquad \text{Eq 3}$$

Where $p_i$ is the oxygen partial pressure [bar] inside the tube, $p_a$ is the ambient oxygen partial pressure [bar] in the room, $p_0$ is the initial oxygen partial pressure in the reactor at the start of the test [bar], $H$ is the solubility of oxygen in water following Henry's law [mg/(L*bar)], $V$ is the volume of the tube [L], $t$ is elapsed time since the start of the test [minutes], and $r$ is the oxygen permeability of the reactor [mg/(minute*bar)]. Note that this is slightly different from a classical permeability, because it does not include thickness and surface area terms. This is because we are specifically interested in the diffusion performance of the tube as a whole system, rather than the individual components. The data collected in the diffusion experiments were fit to Eq 3 using the scipy.optimize python package, allowing $r$, $p_0$, and $H$ to vary. $p_0$ and $H$ were allowed to vary to

prevent the first data point and any small variations in room temperature and ambient pressure from unnecessarily impacting the curve fit. 3 plastic and 3 glass tubes were tested while actively rolling on the rollers, while another 2 (plastic) and 3 (glass) tubes were tested sitting still on the deactivated rollers. A 2-sample t-test was used to determine if the rolled tests and stationary tests for each reactor type yielded significantly different permeabilities. These tests did not reveal significant differences between rolled and non-rolled tests (p = 0.35 for plastic tubes and p = 0.24 for glass). Thus, the rolled and stationary test results were grouped for each reactor type, and only stationary tests were carried out for the double-ended glass VOA vials.

**2.5.5 Limit of detection.**   To determine the limit of detection (LOD) and limit of quantification (LOQ), the 0-1LPM air mass flow controller was replaced with a 0-200ml/min unit to enable better flow control at lower flow rates. The 0-1LPM nitrogen mass flow controller remained in place. A calibration curve was collected from 0.4 to 9.1% oxygen saturation at steps of ~0.2% saturation. These measurements were all performed in triplicate.

The limit of detection and limit of quantification were determined using Eqs 4 and 5 [47]:
Detection Limit:

$$LOD = \left| \frac{3.3\sigma}{m} \right| \qquad\qquad Eq\ 4$$

Limit of Quantification:

$$LOQ = \left| \frac{10\sigma}{m} \right| \qquad\qquad Eq\ 5$$

Where σ is the standard deviation of the 12 repeated measurements at 3.3% oxygen saturation and $m$ represents the slope of the linear section of the calibration curve.

**2.5.6 Variability between cameras.**   Two identical cameras with their internal infrared-blocking filters removed were used to image the same three optode disks to assess the feasibility of using different cameras for calibration and measurement. These cameras were arranged side-by-side at the same distance from the optode disks. The Av and Tv values for both cameras were set to the same values (Av = ƒ5.6, Tv = 0"4).

**2.5.7 Internal infrared-blocking Filter.**   We compare results from one camera with an infrared-blocking filter to results form an otherwise-identical camera without the filter. These response curves were compared to a calibration curve collected using the standard camera system. These tests were primarily carried out to assess whether cost savings are possible by eliminating the infrared-blocking filter removal step.

**2.5.8 Direct-coated vials.**   As an alternative to placing the optode in the vial cap, vials were prepared with the optode dye cocktail applied directly to the inside surface using a small swab to "paint" the optode onto the vial wall. This was carried out with both glass and plastic vials. We also "roughed up" the inside surface of a set of glass vials using a small sanding drum mounted on an electric drill prior to applying the optode coating. These vials were mounted on a 3D-printed manifold to allow them to interface with the calibration system.

# 3 Results and discussion

## 3.1 Optode response testing

**3.1.1 Ambient lighting.**   The results of these comparison tests indicate that, in general, ambient artificial room lighting does not strongly impact the performance of the sensors. Calibration curves for both scenarios produced very high (>0.98) $R^2$ values (Fig 2), and the calibrations are not significantly different from each other. The small difference between

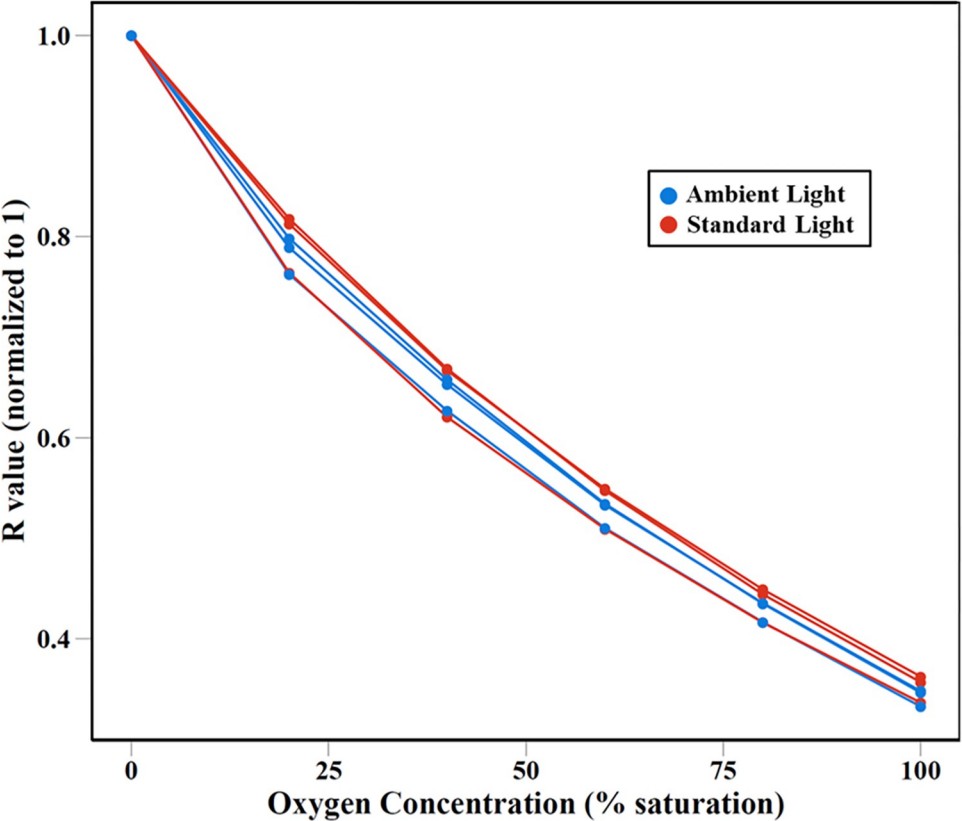

**Fig 2. Calculated R values measured under ambient room lighting conditions and standard LED lighting.**
Standard Light represents calibration curves generated from data collected in the dark room with LED illumination only. Ambient Light represents calibration curves collected under ambient lighting conditions. R values were normalized to a maximum value of 1.

calibration curves with and without ambient indoor lighting indicates that whatever lighting situation is chosen for an experiment, it should be consistent throughout the experiment. It should be noted that only one type of ambient indoor lighting was evaluated here, and the results may not be generalizable across different types of ambient room lighting. The observed variation also indicates that natural light (sunlight rather than ambient artificial light) may be problematic, as it is inherently highly variable.

**3.1.2 Excitation light angle.** A 30˚ angle resulted in a 3.9% increase in indicated oxygen measurement and the 45˚ angle resulted in a 3.8% increase in indicated oxygen measurement (Fig 3). The calibrations are not significantly different from the 0˚ calibration.

The excitation light angle tests indicate that the angle of the excitation light is not very important to the experimental setup, and that the impact of the angle varies in relation to the angle itself. This is convenient, as replicating exact lighting angles between calibration and experiment can be challenging. It also allows for minor repositioning of the lighting system or camera while carrying out the experiment (e.g., from accidental movements or to access parts of the system). Because both of these challenges only result in very small angle changes, it is expected that they will not strongly impact data collected.

**3.1.3 Sterilization: Autoclave and high-pH testing.** Neither the gravimetric nor liquid autoclave cycles resulted in significant visible changes to the optode disks. Under standard lighting, the autoclaved disks looked like disks that had not been autoclaved, however,

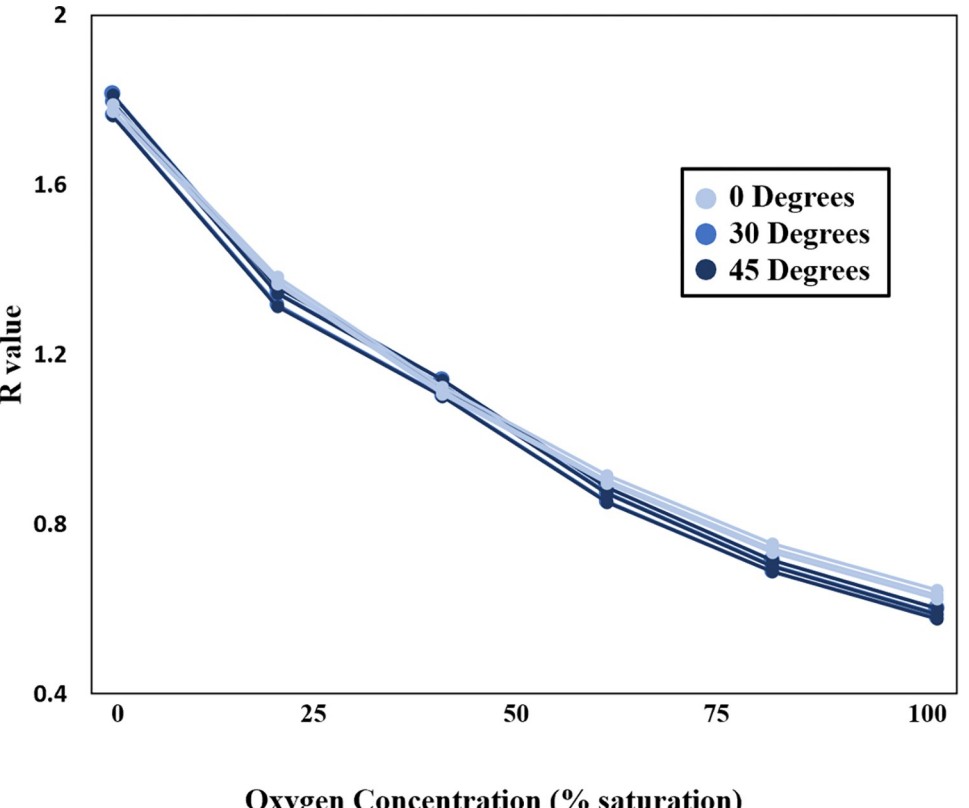

**Fig 3. Measured R value at varied incident light angles.** 0˚, 30˚, and 45˚.

subjecting the two disks to blue lighting shows a distinctly visible difference, with blue-green fluorescence apparent in the autoclaved disk. This may suggest that the indicator dye has broken down and is no longer absorbing the light from the antennae dye (Fig 4). In addition to this visible change, the optode response changed greatly (and significantly) from the original calibration (Fig 4).

The results of the high-pH longevity test (Table 1) show that after 28 days, the optodes have no change in Ksv. The three disks showed an average decrease in R0 of 7%. Despite this, the calibrations carried out at 7, 14, and 28 days were not significantly different from the initial T0 calibration.

In general, autoclaving caused large changes in the performance of the optodes, regardless of the autoclave process chosen. This indicates that autoclaving is not a suggested method for sterilization of the optode disks. The high-pH results indicate that the disks are impacted by long-term exposure to high-pH solutions, however high-pH sterilization, especially with short exposure time, is a better sterilization option than autoclaving. It is recommended to recalibrate the disks after sterilization, in sterile water, if possible.

**3.1.4 Diffusion.** Table 2 includes both the permeability values generated during the diffusion tests, as well as a worst-case-scenario rate of oxygen diffusion assuming 0% oxygen saturation inside the reactor, as this value is helpful in framing the permeability results in a practically useful way.

In general, Table 2 shows that oxygen permeability into the reactors is low relative to many of the oxygen consumption rates reported in environmental studies, and in many experimental setups dealing primarily with higher oxygen concentrations, faster consumption rates, and/

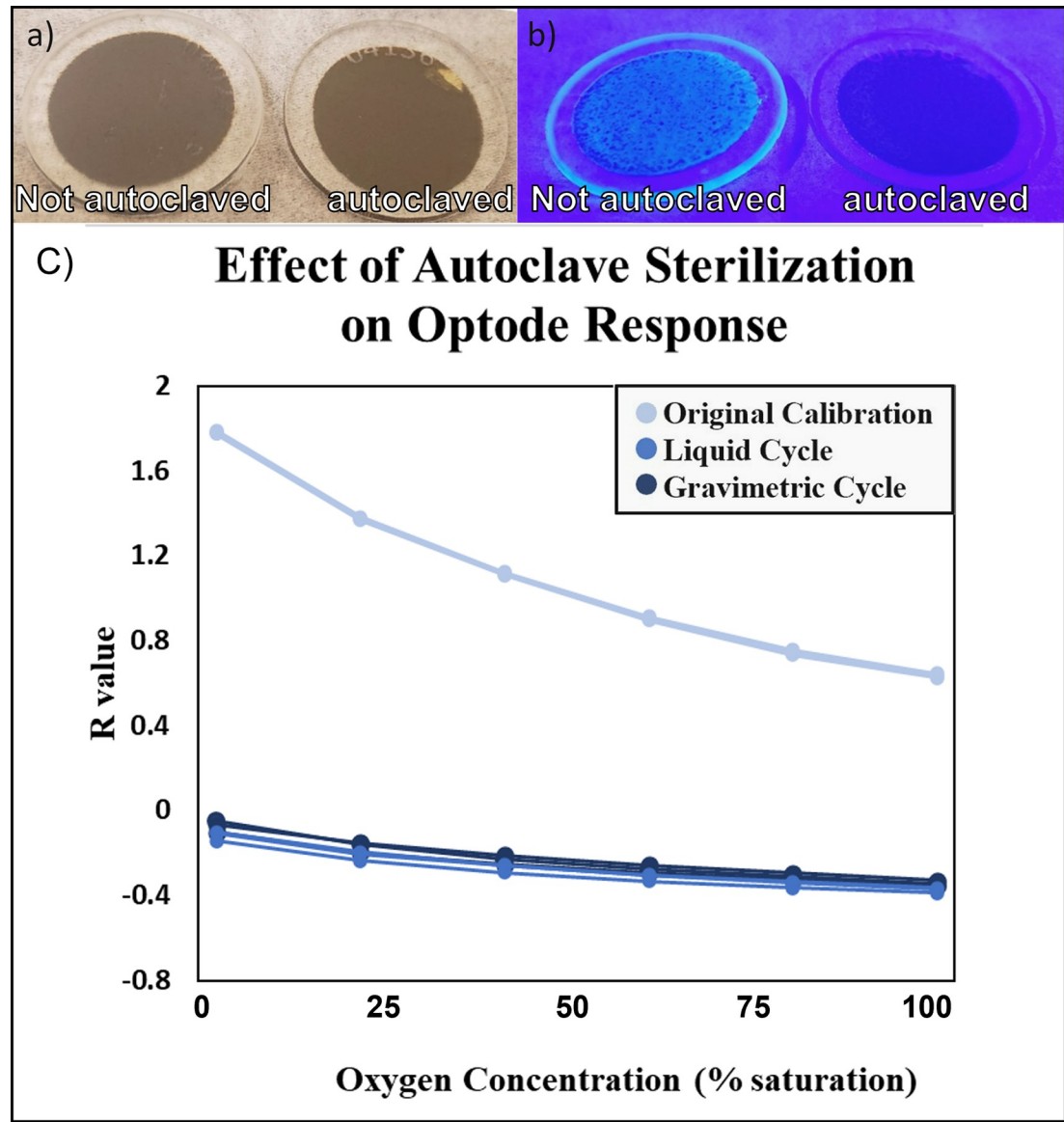

**Fig 4.** (a) Autoclaved optode disk (left) vs. non-autoclaved optode disk (right) under standard lighting. (b) Autoclaved optode disk (left) vs. non-autoclaved optode disk (right) under blue lighting. (c) Results of autoclave sterilization on optode response. Measured R values for original calibration as well as samples autoclaved using the gravimetric 30 and liquid 45 autoclave settings.

**Table 1. Calibration coefficients over time.**

| time(days) | calibration coefficients | | | | | |
| --- | --- | --- | --- | --- | --- | --- |
| | ksv a | ksv b | ksv c | R0 a | R0 b | R0 c |
| 0 | 0.012 | 0.009 | 0.01 | 1.881 | 1.948 | 1.93 |
| 7 | 0.011 | 0.01 | 0.011 | 1.776 | 1.875 | 1.877 |
| 14 | 0.011 | 0.01 | 0.011 | 1.727 | 1.839 | 1.825 |
| 28 | 0.011 | 0.01 | 0.01 | 1.722 | 1.828 | 1.805 |

**Table 2. Diffusion results.**

| reactor type | reactor oxygen permeability (mg/(min*bar)) | worst-case oxygen concentration change rate (mg/(l*hr)) | number of tests - |
|---|---|---|---|
| plastic centrifuge tube | 3.02E-03 | 0.60 | 5 |
| glass voa vial | 1.75E-03 | 0.35 | 6 |
| double-ended glass voa vial | 9.44E-04 | 0.19 | 2 |

or shorter experiment times it may be safe to ignore entirely. This is less likely to be the case in systems with very low oxygen concentrations and/or very low oxygen consumption rates. Because the actual rate of oxygen entering the reactor is dependent on the difference in oxygen partial pressures between the ambient air and the inside of the reactor, ignoring diffusion will tend to bias results towards inferring lower oxygen consumption rates than are actually occurring. Not surprisingly, the diffusion rates are lower in the glass vials than the plastic vials, as the gas permeability of glass is lower than plastic, and also the walls of the plastic reactors are thinner than the glass walls. It may be possible to reduce diffusion further through a combination of thicker optode disks and changing the material chosen for the sealing O-rings (e.g., Viton), as the O-rings used in these tests are made of silicone, which typically has very high gas permeability [48]. Diffusion into the reactors is also generally easy to measure (as described in section 2.5.4), and consistent across replicates of each type of reactor. These aspects allow the effects of diffusion to be accounted for during data processing if desired.

**3.1.5 Limit of detection.** Fig 5 shows the calibration curve collected for calculating the limit of detection and quantification for one of the optode sensors (Eqs 4 and 5). To calculate the limit of detection, the calibration curve was limited to the linear dynamic range, which was estimated to be approximately 0.4 to 40% oxygen saturation ($R^2$ = 0.976; Fig 5). Limits of detection were calculated for each individual optode and averaged to determine the overall limit of detection (LOD); these values were calculated to be 1.15%, 0.58%, and 1.61% oxygen saturation resulting in an average LOD of 1.1±0.4% oxygen saturation. Limit of quantification (LOQ) values were calculated in the same manner as LOD resulting in LOQ values of 3.50%, 1.75%, and 4.88% oxygen saturation with an average LOQ of 3±1% oxygen saturation. These values appear to be conservative, and additional low-oxygen performance may be achievable on an experiment-specific basis.

Limit of detection and limit of quantification tests indicate that this system is capable of performing adequately down to at least 3% oxygen saturation conditions (approximately 0.23 mg/l at 21°C and 1013 millibar). If the experimental needs require high precision at concentrations lower than that, it may be possible to achieve acceptable performance by varying the camera settings (e.g., letting less light into the camera), or possibly by altering the ratio of the dyes in the sensor cocktail. Additional low-concentration performance is likely available either by using the Click-chemistry dye process referred to in the methods section [43], or by coating the dye cocktails onto a thin mylar foil and securing that between the O-ring and the acrylic disk, rather than coating the disk directly. Further experimentation would be necessary in each case.

**3.1.6 Variation between cameras.** Calibration data was collected for three optode caps imaged with each of a pair of separate cameras. After processing the data for both calibration curves for all three optode caps, theoretical R values ranging from 0.4 to 1.5 were inserted into to these calibration curves, generating synthetic data to observe the error that would be induced by switching cameras (Fig 6, left). The synthetic results from the pair of cameras were plotted against each other (Fig 6, right). This shows the deviation from exactly duplicated results, which would be indicated by a slope of 1. The slopes range from 1.02 to 1.04, indicating

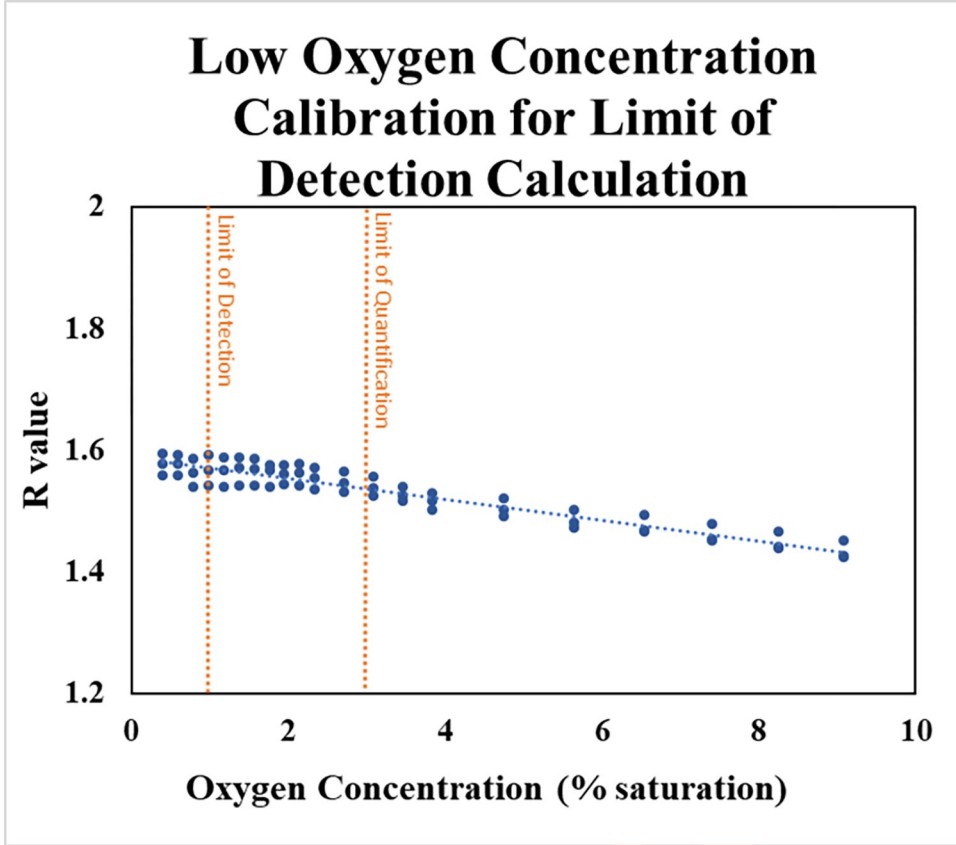

**Fig 5. Limits of detection and quantification estimated from averaged LOD and LOQ calculated for individual optode disks.**

that the 2nd camera reports approximately 4% lower values than the original camera when the same calibration factors are applied to both. The calibrations are not significantly different from eachother.

The variation between cameras shown by these tests is small, on the order of 4% and the calibrations are not significantly different. As a result, it is likely to be generally acceptable to use more than one camera in a given experiment or calibration without performing camera-specific calibration tests. That is, one camera may be set up for calibration while another is used for the experiment. Alternately, if a camera suffers a malfunction during an experiment, it could be replaced with another without strongly impacting the data collected.

**3.1.7 Camera internal infrared-blocking filter.** Traditionally, cameras used for imaging this optode dye cocktail have their internal infrared-blocking filters removed [41]. This process is difficult to carry out without damaging the camera, so the usual path is to send the camera out to a camera shop, which in the authors' experience costs about $200. We assessed how necessary this process is. The results in this section only apply for the specific dye cocktail used in these experiments PtTFPP and MY). Changing the dye cocktail will necessarily change the emission spectra of the dyes, and these results will not be applicable in such a situation. In general, removal of the IR filter decreased the raw amplitude of the red emission light signal vs. a camera where the IR filter was removed, and a direct comparison between with and without-filter calibration curves shows significant difference between them. This is not surprising, as the emission spectrum for the optode dye extends into the near-IR. This decrease in amplitude

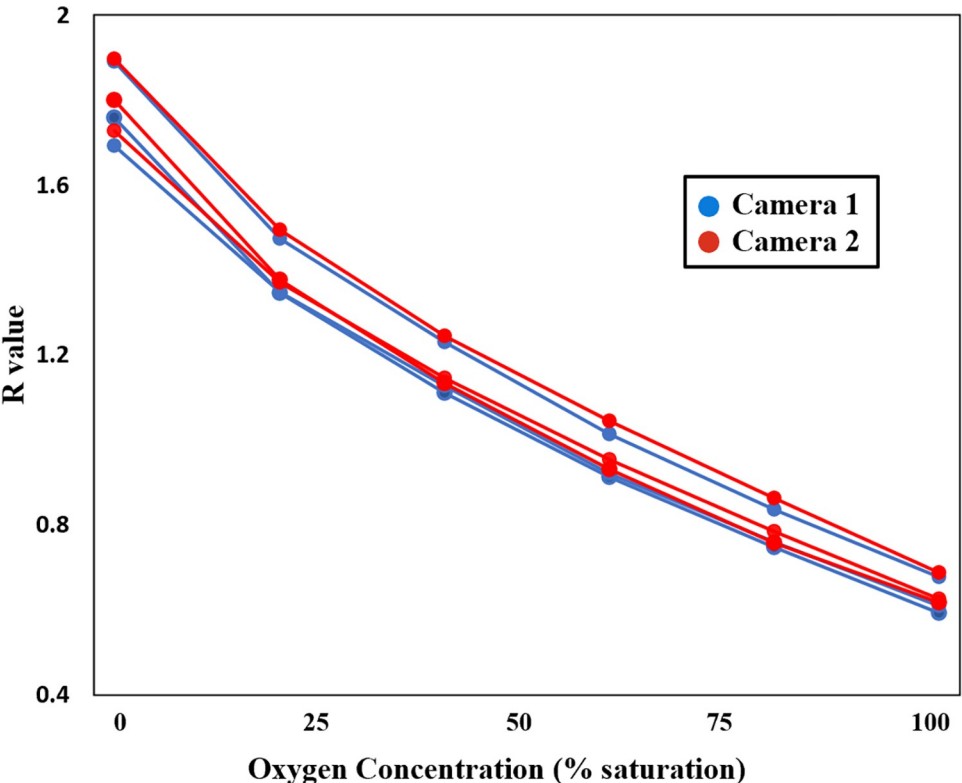

**Fig 6. Calculated oxygen (percent saturation) values calculated from sequence of R values.** Camera 1 is the original camera used for previous measurements, while Camera 2 is a second, identical camera.

can be compensated for by increasing the excitation light brightness, the exposure time, or some combination of both. One consideration is that the equation used by Larsen, Borisov [41] to calculate R returns negative values when the green amplitude exceeds the red amplitude, and this causes curve-fit failures using the modified Stern-Volmer equation they present for calibration. Replacing the Larsen R with a simple ratio of red amplitude to green amplitude solves this issue. When this approach is followed, calibration curves with similar $R^2$ values are produced with and without the internal IR-blocking filter (Fig 7). This means that users can elect not to have that filter removed, as long as the additional excitation light brightness and/or longer exposure time are acceptable. This represents a cost savings, as well as avoiding the logistical complication of having the camera modified by a professional camera technician. It is not recommended to switch between cameras with/without filters without recalibrating.

### 3.2 Direct-coated vials

Successfully direct-coating of the inside surface of the vials proved challenging. Both plastic and glass vials suffered from nearly-immediate detachment of the cured optode from the inside surface of the vial when the coated vial was filled with water. The "roughed up" glass vials performed better, with the optode staying adhered to the vial walls throughout calibration and testing. However, even with these vials, the optodes eventually detached from the walls after several days of immersion in water.

Calibration curve-fits of the direct-coated vial were very similar to the standard optode disk. This suggests that the direct-coated vials are comparable in function and could be used as an alternative approach for these analyses. The direct-coated vials do suffer from the dye

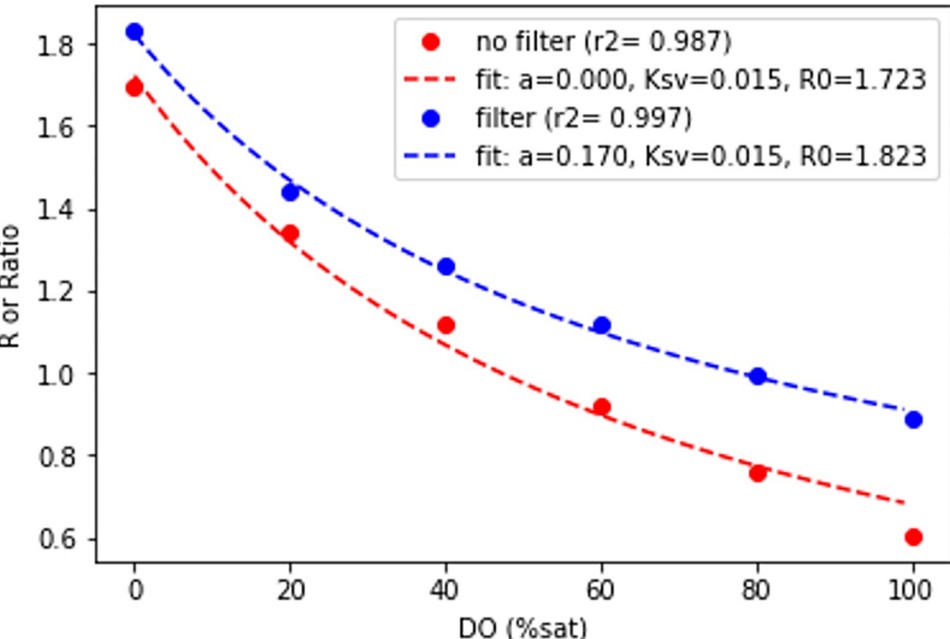

**Fig 7. Comparison of calibration curves between a camera with the internal IR-blocking filter removed (red) and intact (blue).** In the case of the removed IR filter, R was calculated according to Larsen, Borisov [41]. In the case of the intact IR filter, the simple ratio of red amplitude to green amplitude was used.

cocktail detaching from the glass vial surface after prolonged immersion in water, perhaps due to the polystyrene absorbing water and swelling, since the adhesion between the polystyrene matrix and the glass vial is much weaker than between the polystyrene and the acrylic disk. It is possible that using acrylic (PMMA) or polystyrene vials would alleviate the adhesion issues. The glass direct-coated vials could nonetheless be of use in a situation where the lid of the vial cannot be modified to work with the optode disk, such as when a pierceable septum or a crimp-sealed bottle top is needed, or when it is highly desirable to image the vial from the side, perhaps in a situation where there is sediment at the bottom of the vial, and a supernatant liquid above.

## 4. Multireactor advantages

There are several areas where the multireactor compliments or provides a potentially useful alternative to existing systems. These fall into three primary areas: reduced cost and labor for large numbers of data points, high temporal data resolution, and customizability/flexibility.

With regard to cost and labor, the parts cost to assemble the multireactor system itself add up to approximately $3000. Each additional roller rack costs roughly $1000. The materials cost to produce the optode disks varies with the number produced, but the supplies needed to make 500 disks cost less than $2000, assuming the user can find access to a laser cutter and a fume hood, common items at many research institutions. The calibration system is a significant expense, with parts totaling about $7000. The initial cost is somewhat high compared to individual commercial sensor options. If all the researcher needs is a start and end concentration, it would generally be cheaper to use a commercially-available system such as the Fibox3 (PreSens Precision Sensing GmbH, Regensburg, Germany) or Firesting GO2 (Pyroscience, Aachen, Germany) along with the associated contactless sensor spots. Multi-point versions of such systems can easily cost well over $10,000 however, and it is worth noting that the cost for

the multireactor does not scale linearly with the number of reactors or time-points. The imaging and calibration systems are one-time expenses, and the disks can be produced in large numbers for little extra cost. The only significant parts cost of increasing scale is the tube rollers, which are a similar expense regardless of the oxygen measurement system used. With that in mind, where this system really shines in terms of reduced labor costs.

In preliminary experiments, in a 4 hour timespan we generally can collect data every 2 minutes from 20 reactors for 2 hours, with a total labor expenditure of 4.5 person-hours for preparation, 0.5 person-hours for monitoring the system, and 1 person-hour for data extraction. This adds up to approximately 1200 data points, and 0.005 person-hours per data point. It is difficult to compare this to existing methods from literature, as the labor effort is rarely explicitly discussed, however we can look at total measurement numbers. Thamdrup, Hansen [21] analyzed samples from 2 collection dates across 52 temperatures with 3 to 4 oxygen concentration readings per incubation, for a total of some 364 data points. Van Dael, De Cooman [19] carried out 30 incubations with 6 oxygen concentration readings per incubation for a total of 180 data points. Von Schiller, Datry [20] carried out 200 incubations, with only one end-time oxygen concentration reading for each. From personal communications with Garayburu-Caruso, Stegen [18], the labor required to produce one datapoint per reactor per hour for 6 hours with 32 to 35 tubes in a single work day was 2 person-hours to get the reactors started and 10 person-hours to monitor and record the oxygen data. This equates to approximately 200 data points and 0.06 person-hours per data point. While these are rough approximations, the multireactor system can generate ~10x more data in a span of 4 hours than is commonly used for entire studies/publications. The multireactor can also provide ~10x more data produced per unit of labor. These are significant practical aspects that could enable experiments that could not previously be completed due to logistical constraints associated with data production rates and costs.

The ability to collect high time-resolution data is also advantageous in many situations. As discussed in the prior paragraph, many existing studies collect between one and 6 data points per incubation, over times ranging from hours to days. Collecting a small number of data points over the course of an incubation forces the researcher to make assumptions about the shape of the curve of oxygen concentration over time. This is particularly true in the case of 1 or 2 data points, as these inherently produce a straight line. It is also important in situations where the reactor reaches anoxic conditions, since with few data points it is impossible to tell when anoxic conditions were reached, and thus the rate of oxygen consumption is poorly determined. In many cases, the shape of the curve can also offer information about the potential processes occurring in the reactor.

Fig 8 shows one example incubation of river sediment in water over 70 minutes, with data recorded every 2 minutes. The first data point is well under air-saturated conditions, indicating that the time it took to prepare the sample (a few minutes) was enough for the system to consume some oxygen. The first ~10 minutes show nonlinear decline in oxygen, which could be indicative of a first-order-to-oxygen reaction. After that, the decay in oxygen becomes linear, approximating a 0-order reaction, and finally at ~42 minutes, the system reaches anoxic (or at least very low oxygen) conditions. It is possible that what is being observed here is an initial combination of biotic and abiotic oxygen consumption, with abiotic processes tending toward 1$^{st}$-order oxygen decline, followed by the cessation of abiotic oxygen consumption and the continuation of non-oxygen-limited biological processes which show up as a 0-order linear decay, and finally achieving an oxygen-limited phase. None of this would be apparent with only a pair of data point at time 0 and time 60. Higher temporal resolution also allows for the opportunity to carry out more robust time-series outlier detection and removal than is possible with only a few data points, which gives the researcher more ability to trust their results.

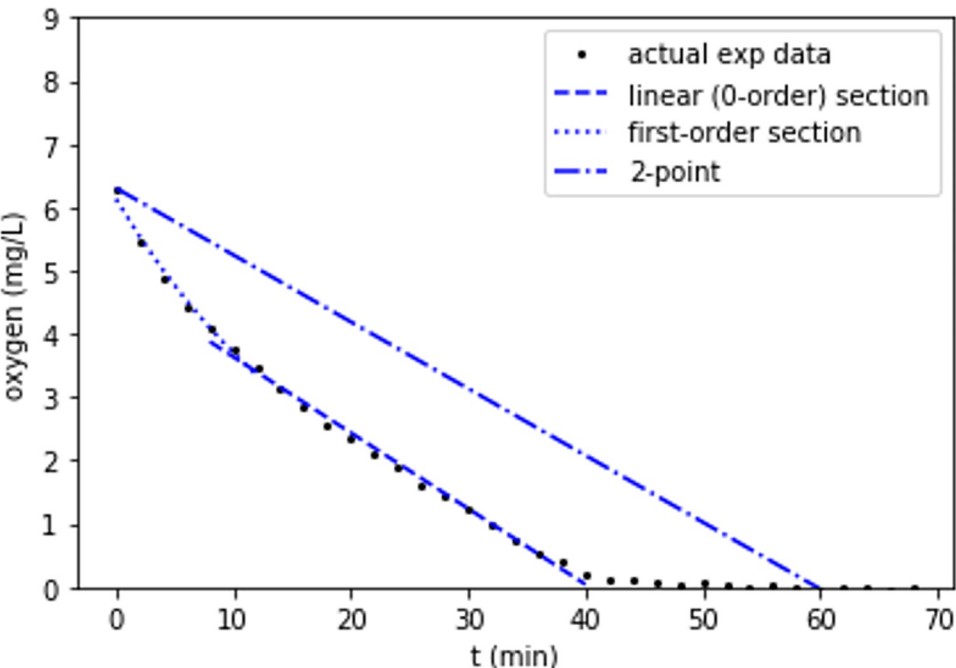

**Fig 8. Oxygen concentration in a small reactor filled with water and river sediment.** Data was collected every 2 minutes for 70 minutes. A period of nonlinear decline in oxygen is followed by a period of linear decline, and finally anoxic conditions are achieved. None of these details are visible in the theoretical example of a 2-point experiment with reads at 0 and 60 minutes.

Developing automated real-time image and data analysis would further allow for sampling specific multireactor vials at specific oxygen concentrations (e.g., during the oxic-to-anoxic transition). Additionally, higher-frequency measurements can be leveraged to provide more accurate consumption rate measurements, as described by Holtappels, Tiano [49].

Another useful aspect of the multireactor system is its flexibility and adaptability. While oxygen consumption is important, it is rarely the only process of interest in a modern metabolic experiment. The photograph-based data collection method used by the multireactor allows for a lot of leeway in what exactly the target reactors consist of. The size and material of the reactors are adjustable to allow for such considerations as RNase-free vials to allow genomic analyses post-incubation, or glass vials for minimal gas permeability and reactivity, and even PEEK vials with thicker acrylic caps to allow for above-atmospheric-pressure incubations. Double-ended glass vials can be purchased, which allows the use of an optode disk in one and while maintaining a pierceable septum in the other, for headspace gas and/or liquid samples to be taken either during or after the incubation. In general, if the reactors can be arranged in front of the camera, the system can accept almost any configuration. Detailed understanding of biotic and abiotic rates and processes can be achieved by using different configurations across experiments. Additionally, because the disks are imaged at relatively high spatial resolution, it is possible to examine the results and assess heterogeneity within each sample.

## 5. Summary, limitations, and next steps

There is a need to understand impacts of disturbance and changing climate conditions on many ecosystems, ranging from soils to freshwater to marine, including both open water and sediments/soils. To generate mechanistic knowledge of how these systems operate and

respond to disturbance requires manipulative experiments. Manipulative experiments require a large number of data points including both experimental treatments and replication. They also require a flexible system that can be modified to provide multiple data types. The metabolic multireactor provides the inexpensive, high throughput, flexible basis required to enable these studies.

To summarize, key practical considerations for using the system include:

- Ambient lighting–ambient lighting consistency is important. If artificial indoor ambient fluorescent tube light (as is common in lab spaces) is held constant, it is not a problem.

- Excitation light angle is forgiving. Small changes in excitation light angle do not cause substantive changes in optode response.

- Autoclave sterilization is not recommended. We recommend NaOH for sterilization, as high-pH stability of the optodes is relatively good.

- Diffusion of oxygen into the reactors from ambient air is likely not a problem for higher-oxygen concentration and/or shorter (1–2 hours) studies. The impact of diffusion can be measured and compensated for if necessary.

- The limit of detection is 1.1% air saturation and the limit of quantification is 3% (0.1 and 0.27 mg/l respectively, at 21°C and 1013 millibar). If best performance is required at oxygen concentrations below 10% saturation (0.9 mg/l), it is recommended to carry out a low-concentration calibration.

- The difference in measured concentration between identically-prepared cameras is small. It is therefore acceptable to use multiple cameras within a set of calibrations and experiments.

- IR blocking filter–the IR-blocking filter can remain in place without strongly impacting system performance, as long as the dyes used are not changed, and a simple ratio of red intensity to green intensity is used in place of [41]'s R value.

- Direct-coated vials with smooth walls did not provide sufficient adhesion between the optode and the vial wall. The "roughed up" glass vials performed well for oxygen, but the roughness introduces potential inconsistencies, and was still not as durable as a tube-cap optode. If a pierceable septum is required for additional data types, the double-ended glass vial is an attractive (if somewhat expensive) solution.

While the system provides a great deal of flexibility, control, and throughput, there are some significant limitations and opportunities for improvement. More and different tests of system performance are also always desirable, as time and funding allow. As it stands, the system does not actively monitor the temperature of the tubes. As the calibration is temperature dependent, we currently rely on making sure that the room temperature is well controlled and that the tubes are in thermal equilibrium with the room air. A great addition to this system would be infrared monitoring of the tube temperatures, either through a 2-d infrared camera, or an array of single-point infrared non-contact temperature sensors. This would allow for more complex studies that involve temperature changes, and prevent errors due to insufficiently well-controlled room temperature or equilibrated tubes. Also of great value would be improved data processing workflows. The existing process relies on substantial manual record keeping and data extraction using ImageJ software, both for calibrations and experiments. An integrated experiment tracking and data processing software package would offer significant improvements in ease of use and labor. Finally, this system has only been tested with relatively small bioreactors in a horizontal-roller system, using one type of camera. The roller system is

configurable to use much larger (up to several liters) bottles, and it may be possible to position the camera and lights above a vertical laboratory shaker. Evaluation of the differences between camera manufacturers and models may also be useful. The opportunities for customizing the base system are quite broad.

Overall, this is a robust, relatively inexpensive, and highly flexible system that allows for high-throughput batch reactor studies and is particularly suited to manipulative experiments with wide parameter space. Pre- and post-incubation analyses of the gas, water, and sediment contained in the reactor can be easily paired with the oxygen consumption data, helping to link metabolic rates with changes in pH, carbon dioxide, nitrogen species, microbial communities, organic matter transformations, and many other data types. As such, this system can provide a base for broad manipulative studies across ecosystems. Some examples include examining the aerobic respiration response to varying pH and salinity in systems ranging from upland freshwater sediments through tidal and into marine, or varying abundance of nutrients, organic carbon, and even contaminants. With this system we hope to provide an approachable common platform to generate transferrable knowledge across a wide range of ecosystems and perturbations.

## Supporting information

**S1 File. 3d printer and CAD files for various 3d printed calibration fixtures.**
(SCAD)

**S2 File. 3d printer and CAD files for various 3d printed calibration fixtures.**
(STL)

**S3 File. 3d printer and CAD files for various 3d printed calibration fixtures.**
(SCAD)

**S4 File. 3d printer and CAD files for various 3d printed calibration fixtures.**
(STL)

**S5 File. 3d printer and CAD files for various 3d printed calibration fixtures.**
(SCAD)

**S6 File. 3d printer and CAD files for various 3d printed calibration fixtures.**
(STL)

**S7 File. 3d printer and CAD files for various 3d printed calibration fixtures.**
(SCAD)

**S8 File. 3d printer and CAD files for various 3d printed calibration fixtures.**
(STL)

**S9 File. 3d printer and CAD files for various 3d printed calibration fixtures.**
(SCAD)

**S10 File. 3d printer and CAD files for various 3d printed calibration fixtures.**
(STL)

**S11 File. 3d printer and CAD files for various 3d printed calibration fixtures.**
(SCAD)

**S12 File. 3d printer and CAD files for various 3d printed calibration fixtures.**
(STL)

**S13 File. 3d printer and CAD files for various 3d printed calibration fixtures.**
(SCAD)

**S14 File. 3d printer and CAD files for various 3d printed calibration fixtures.**
(STL)

**S15 File. 3d printer and CAD files for various 3d printed calibration fixtures.**
(SCAD)

**S16 File. 3d printer and CAD files for various 3d printed calibration fixtures.**
(STL)

**S17 File. Oxygen saturation vs. R values with temperature varied from 9.0˚C to 29.5˚C.**
Temperature was increased and decreased by immersing the water reservoir in a water heater
or an ice bath, respectively.
(DOCX)

**S18 File. Plot depicting potential error ranges for uncorrected temperature effects (in systems ranging from 9˚C to 30˚C.** The dashed line approximates room temperature, the upper
line indicates measurements made at 9.0˚C, and the lower line indicates measurements made
at 30˚C. The x-axis represents actual oxygen measurement values under ambient temperatures
(~19˚C) and the y-axis represents the resulting measurements at varied temperature.
(DOCX)

**S19 File. Salinity values were varied by adding NaCl to the reservoir at concentrations
ranging from 0% to 3.5% (m/v).** Oxygen saturation values were calculated using the equation
$DO_{salt} = DO - qS$ where $DO_{salt}$ is the dissolved oxygen concentration (in mg/L) of air-saturated
salt water, and DO is the dissolved oxygen concentration (mg/L) of air-saturated distilled
water, and q is approximated using the equation: $q = -0.1903t + 12.892$, where t is temperature
in degrees Celsius.
(DOCX)

**S20 File. pH values were increased and decreased by adding sodium hydroxide and hydrochloric acid, respectively, to vials containing varied concentrations of oxygen.** The values in
the plot above have been adjusted to account for differences in temperature between vials.
Overall, pH was not observed to significantly influence optode response (outside the ±3%
range) except at pH below ~4.5.
(DOCX)

**S21 File. Optode response in the presence of nitrate was tested by amending the reservoir
solution with sodium nitrate at a concentration of 1,000mg/L.** No significant deviation
from results obtained at 0mg/L nitrate were observed.
(DOCX)

## Acknowledgments

In no particular order, the authors would like to thank Dillman Delgado, Maggi Lan, Kenton
Rod, Lupita Renteria, Sophia McKever, Aditi Sengupta, Vanessa Garayburu-Caruso, Swatantar
Kumar, the Pacific Northwest National Laboratory Instrument Development Laboratory,
Morten Larsen, Adam Kessler, and the many other individuals who contributed ideas, methods, testing, and data collection either for this system directly, or in projects that lead us here.

## Author Contributions

**Conceptualization:** Matthew H. Kaufman, James C. Stegen.

**Data curation:** Matthew H. Kaufman, Joshua Torgeson.

**Formal analysis:** Matthew H. Kaufman, Joshua Torgeson.

**Funding acquisition:** Matthew H. Kaufman, James C. Stegen.

**Investigation:** Matthew H. Kaufman, Joshua Torgeson.

**Methodology:** Matthew H. Kaufman, Joshua Torgeson, James C. Stegen.

**Project administration:** Matthew H. Kaufman.

**Resources:** Matthew H. Kaufman, James C. Stegen.

**Software:** Matthew H. Kaufman, Joshua Torgeson.

**Supervision:** Matthew H. Kaufman.

**Validation:** Matthew H. Kaufman.

**Visualization:** Matthew H. Kaufman, Joshua Torgeson.

**Writing – original draft:** Matthew H. Kaufman, Joshua Torgeson.

**Writing – review & editing:** Matthew H. Kaufman, Joshua Torgeson, James C. Stegen.

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
