## [Decision Letter · Decision Letter 0]

3 May 2023

PONE-D-23-06927Metabolic Multireactor: practical considerations for using simple oxygen sensing optodes for high-throughput batch reactor metabolism experimentsPLOS ONE

Dear Dr. Kaufman,

Thank you for submitting your manuscript to PLOS ONE. After careful consideration, we feel that it has merit but does not fully meet PLOS ONE’s publication criteria as it currently stands. Therefore, we invite you to submit a revised version of the manuscript that addresses the points raised during the review process.

 The manuscript presents an alternative incubation setup that enables to measure the oxygen consumption in multiple reactors simultaneously. Overall, the study is a valuable contribution to the field and it has potential for wide applicability in other fields. Nevertheless, the authors are encouraged to improve the manuscript following the reviewers recommendations which will help  to improve the quality and significance of the work presented.

We look forward to receiving your revised manuscript.

Kind regards,

Catarina Leite Amorim, Ph.D.

Academic Editor

PLOS ONE

Journal Requirements:

3. Please expand the acronym “PNNL LDRD” (as indicated in your financial disclosure) so that it states the name of your funders in full.

"..A portion of the research described in this paper was conducted under the Laboratory Directed Research and Development Program at Pacific Northwest National Laboratory, a multi-program national laboratory operated by Battelle for the U.S. Department of Energy. MK was grateful for the support of the Linus Pauling Distinguished Postdoctoral Fellowship program. This research was supported by the U.S. Department of Energy, Office of Science, Office of Biological and Environmental Research, Environmental System Science (ESS) Program. This contribution originates from the River Corridor Scientific Focus Area (SFA) project at Pacific Northwest National Laboratory (PNNL). This research was supported under award DESC0018042."

"A portion of the research described in this paper was conducted under the Laboratory Directed Research and Development Program at Pacific Northwest National Laboratory, a multi-program national laboratory operated by Battelle for the U.S. Department of Energy. MK was grateful for the support of the Linus Pauling Distinguished Postdoctoral Fellowship program. This research was supported by the U.S. Department of Energy, Office of Science, Office of Biological and Environmental Research, Environmental System Science (ESS) Program. This contribution originates from the River Corridor Scientific Focus Area (SFA) project at Pacific Northwest National Laboratory (PNNL). This research was supported under award DESC0018042.

The sponsors and funders did not play any role in the study design, data collection and analysis, decision to publish, or preparation of the manuscript."

5. Thank you for stating the following in your Competing Interests section: "No"

7. We note that you have stated that you will provide repository information for your data at acceptance. Should your manuscript be accepted for publication, we will hold it until you provide the relevant accession numbers or DOIs necessary to access your data. If you wish to make changes to your Data Availability statement, please describe these changes in your cover letter and we will update your Data Availability statement to reflect the information you provide.

Reviewers' comments:

Reviewer's Responses to Questions

**Comments to the Author**

1. Is the manuscript technically sound, and do the data support the conclusions?

Reviewer #1: Yes

Reviewer #2: Yes

2. Has the statistical analysis been performed appropriately and rigorously? 

Reviewer #1: Yes

Reviewer #2: No

3. Have the authors made all data underlying the findings in their manuscript fully available?

Reviewer #1: Yes

Reviewer #2: Yes

4. Is the manuscript presented in an intelligible fashion and written in standard English?

Reviewer #1: Yes

Reviewer #2: Yes

5. Review Comments to the Author

Reviewer #1: The manuscript entitled "Metabolic Multireactor: Practical Considerations for Using Simple Oxygen Sensing Optodes for High-throughput Batch Reactor Metabolism Experiments," authored by Kaufman and colleagues, introduces a novel incubation setup that enables simultaneous measurement of oxygen consumption in multiple reactors. The authors conducted meticulous testing of the system and provided comprehensive supplementary information, including all CAD designs. Overall, the study presents a valuable contribution to the field and showcases the potential of this methodology for high-throughput batch reactor metabolism experiments.

The proposed system has the potential for wide applicability in various fields of research, including marine, aquatic, and terrestrial studies. This broad scope of potential applications makes the manuscript an appealing read for a broad readership, and its suitability for publication in PLOS ONE is evident. As the reviewer, I have opted for a "major revision" decision, as I believe there is room for improvement in certain sections of the manuscript. Including these clarifications will improve the overall clarity and understanding of the experimental design.

Reviewer #2: The manuscript by Kaufman et al details their development of a microbioreactor system with integrated oxygen measurements via optodes. The authors describe their system (with appropriate supporting information to enable fabrication) and the extensive calibration and determination of system parameters. Important for a paper of this type, the methods section is very clearly written with a suitable amount of detail - one of the best written methods section I've reviewed recently. Overall, all of the experiments the authors completed to analyze their system are appropriate, and the overall presentation of the work is excellent. While I have some minor concerns and suggestions (detailed below), the manuscript is certainly suitable for publication with some minor changes (the most important being incorporation of statistical tests to assay significance of differences) and I look forward to it's publication.

Minor Issues & Suggestions

- Several sections would benefit from statistical analysis. Basically anywhere where different conditions are being compared (e.g. fig 3, fig 4, fig 5c, possibly fig 6, fig 7, fig 8, some SI figures, and accompanying test. Just doing a comparison and adding if it is significant or not would help to clarify. E.g. - p19 - "reports approximately 4% lower values" but I would guess not significantly different.

- Probably worth changing "fluorescent dye" etc. to luminescent given the mechanism for the dyes

- Possibly worth cutting 2.5.7 and 2.5.8 from the methods section and integrating into the results and discussion. Not much actual "methods" and more explanation as for motivation.

- Figure 5 - possibly add labels to figure (e/g. "autoclaved""not autoclaved" to image) to help make it easier to understand at a glance

- P17 (section 3.1.4) "shows that the oxygen permeability into the reactors is low" - add range for what "low" is or explain to the readers. I.e. maybe worth comparing with normal units and show that it is considered low?

- Figure 6 (and other may benefit too) - smaller data points to show individual points more clearly.

- Section 3.1.5 mentions click chemistry detailed in methods, but I didn't see it there (may have just missed it).

- Probably not worth it, but maybe comparing with a different camera type in addition to a duplicate camera would add value for the reader (i.e. same camera - calibration probably fine, different camera - new calibration needed).

-

6. PLOS authors have the option to publish the peer review history of their article (what does this mean?). If published, this will include your full peer review and any attached files.

Reviewer #1: No

Reviewer #2: No

---

## [Author Response · Author response to Decision Letter 0]

12 Jun 2023

Dear PlosONE staff and reviewers,

We greatly appreciate the time and energy put into the review process for this manuscript. Below please reviewer comments and our responses.

Reviewers' comments:

Reviewer #1: The manuscript entitled "Metabolic Multireactor: Practical Considerations for Using Simple Oxygen Sensing Optodes for High-throughput Batch Reactor Metabolism Experiments," authored by Kaufman and colleagues, introduces a novel incubation setup that enables simultaneous measurement of oxygen consumption in multiple reactors. The authors conducted meticulous testing of the system and provided comprehensive supplementary information, including all CAD designs. Overall, the study presents a valuable contribution to the field and showcases the potential of this methodology for high-throughput batch reactor metabolism experiments.

The proposed system has the potential for wide applicability in various fields of research, including marine, aquatic, and terrestrial studies. This broad scope of potential applications makes the manuscript an appealing read for a broad readership, and its suitability for publication in PLOS ONE is evident. As the reviewer, I have opted for a "major revision" decision, as I believe there is room for improvement in certain sections of the manuscript. Including these clarifications will improve the overall clarity and understanding of the experimental design. 

We thank the reviewer for their in-depth review and constructive criticism.

General comments

1. Heterogeneity measurements: In the introduction, the authors discuss the importance of 

heterogeneity measurements, which I completely agree with. In my opinion, the proposed 

system has an key advantage that is currently not addressed: it can directly resolve 

heterogeneity on the micrometer to millimeter scales. By imaging the roughly 2 cm dye spots, 

the authors can extract information about the variability of the sensor-dye. This approach would 

only require determination of the minimum resolvable spatial resolution.

We have added information on resolution to the end of the imaging system methods section. Also in the “multireactor advantages” section, the ability to asses heterogeneity was added. Digging into specific statistics/methods for this will be a nice topic for a future manuscript.

2. Improve precision through higher temporal resolution: The manuscript does not explore the 

potential benefits of increasing the temporal resolution of the readouts to enhance the accuracy 

of rate measurements. This is an interesting avenue for future research that could be discussed 

or acknowledged within the manuscript. By improving the temporal resolution of the system, 

researchers may be able to obtain more precise measurements of metabolic rates (for 

comparison doi.org/10.1371/journal.pone.0089369 ). 

Additional discussion of this aspect of higher-frequency sampling has been added to the fourth “Multireactor Advantages” paragraph, as has the provided useful reference.

3. IR blocking: For the applied sensor-dye (PtTFPP) the IR blocking is not really important, as most 

of the emission is occurring in the lower red range (600 nm – 680 nm). This might however 

change for other dyes that are utilized by the community. This should be clarified and the

sections on the IR blocking should be toned down. 

We have added the specification that the IR filter section only applies to the specific dye cocktail used in these experiments, and is likely not be applicable to other dyes, to section 3.1.7.

4. Terminology: Throughout the manuscript, the author uses terms such as "metabolism of organic 

material," "microbial metabolism", "aerobic metabolism", "microbial respiration", and 

"consumption" interchangeably, despite the fact that these terms describe different processes. 

While the measurements obtained are primarily oxygen consumption measurements, which

integrate a range of processes, including microbial respiration and oxidation of reduced 

material. It is important to clearly describe this link between organic carbon remineralization 

and oxygen respiration in the introduction and use consistent terminology throughout the 

manuscript. I recommend to use "oxygen consumption," to accurately reflect the measurements 

obtained and avoid confusion.

We have reduced the range of terms used to “oxygen consumption” and “metabolism”. We have also added a few lines in the introduction to discuss the non-metabolic contributors to oxygen consumption rates. “Aerobic metabolism rates are frequently reflected in oxygen consumption rates, although inorganic carbon remineralization and abiotic reoxidation of reduced species can also contribute to oxygen consumption rates.”

5. Calibration: The authors are performing and describing in detail the calibration procedure, 

however the Stern-Volmer equations are missing and should be integrated in the method 

section then it also becomes clearer to which parameter the authors are referring in table 1.

We have added the modified version of the Stern-Volmer equation used by Larsen et. al. [2014] to the methods section (now equation 1), including definitions of the variables referred to in table 1.

 In my opinion the limit of detection (1.1%) is very conservative and it does not reflect the actual 

data (Figure 9, Zoom into 0-0.6 mg/L and add the 1.1% line). 

We have added a statement to section 3.1.5 stating that these limits appear to be conservative, and additional performance may be achievable on an experiment-specific basis.

6. Literature: The reference list is relatively short and does not reflect the breadth of research in 

this field. The proposed system has the potential to be applied in various environments, 

including marine, aquatic, and terrestrial studies. Therefore, it would be beneficial for the 

authors to expand their literature review and include more relevant studies. Additionally, some 

recent novel developments have addressed heterogeneity on the micrometer scale, which 

should also be considered in the manuscript.: doi.org/10.1016/j.crmeth.2022.100216

We have added several references to the introduction, both in the optode section and the existing metabolic research section. There is a huge quantity of literature in these fields, so we are necessarily limited in the scope of what we can site for context here. 

Line and figure specific comments

Figures: To reduce the total number of figures in the manuscript, I recommend combining multiple 

panels into a single figure. Rather than presenting each panel as a separate figure, combining them will 

simplify the presentation and make it easier for readers to follow the results. This approach will also 

reduce the total number of figures, making the manuscript more concise and easier to navigate. Figure 1 and figure 2 can be combined. Add scale bar to figure 1 (panel bottom left).

Figures 1 & 2 have been combined into a single figure. All figure numbers have been updated througout the manuscript.

Abstract: The abstract could be restructured for better clarity. It may be more effective to begin with 

the importance/introduction and limitations of current systems, followed by the setup and advantages 

of the proposed system. Also, it may not be necessary to include the reference to Larsen et al.. Instead, 

it could be stated that ratio-metric sensor-dyes are used, along with the names of the specific dyes. 

We appreciate this comment, however we prefer to keep the abstract ordered the way it is, with the system and its advantages listed first, and without the methodological details of the ratiometric dyes and dye names. We believe that this provides a concise and accurate description of the manuscript, and further details are readily available in the body.

Line 32: See general comment 4 

Please see response to general comment 4.

Line 46: The authors should give examples of such “whole-system approaches” 

Several references to “whole system” approaches are provided at lines 42-43, however we have added a bit more explanation of those types of experiments. “Such studies are often approached either through open-water whole-system approaches based on oxygen time series data…”

Line 52: The authors should describe what is meant with “manipulative experiments” 

We have added a line or 2 of explanation at that location. “(that is, experiments where a system is intentionally subjected to a specific set of conditions, rather than an experiment where field conditions are left largely uncontrolled)”

Line 62: The reference [11] seems to not reflect the statements made. 

We have removed the reference from that location. It was intended as an example of one of the “best existing solutions”, but it was not a great example of the difficulties experienced trying to carry out these kinds of experiments.

Line 69-70: I disagree to the statement that optodes are traditionally “used to obtain two-dimensional 

data”. Traditionally optodes are used in the context of microsensors ( doi.org/10.1016/S0925-

4005(97)80168-2 ) 

We have added some brief discussion of optode microsensors and related studies to this section.

Line 118: Effects of bleaching can also be reduced, calibration is improved, etc. pp. Further, while I agree 

that it is not necessary to provide an in-depth description of the dyes and their chemical properties, it is 

important to mention which reference dye was used in the manuscript. Additionally, the distinction 

between the antenna and reference dyes could be clarified for readers. It is the same dye, but the 

reference dye also serves as an antenna dye, facilitating energy transfer. Also write out once for what 

PtTFPP actually stands for.

We have added the reference/antenna dye name to the methods section. PtTFPP is defined in line 109. We have added a brief statement about the dual role of the Macrolex Yellow GN antenna/reference dye, but we leave most of the details of that system to the cited Larsen paper. 

Line 274: It is rather the “variability” then the “variance” that is testes 

corrected

Line 337: The wording (“Lifetime test”) is a bit confusing as the lifetime of the quenching can also be 

used for calibration. 

This is a good catch. We have changed the term used to “longevity” to prevent confusion.

Line 465: The authors are very conservative with their statements regarding how inexpensive their 

system is. Commercial solutions for similar 4-channel modules are substantially more expensive, such as 

the Pyroscience system, which costs approximately $10,000 and that does not include any of the spots 

and additional lab equipment (and allows only the read-out of 4 spots). It may be worth making stronger 

statements regarding the cost-effectiveness of the proposed system in comparison to existing 

commercial solutions.

We have added a sentence to that section mentioning how expensive commercial multi-point oxygen sensor systems are.

Reviewer #2: The manuscript by Kaufman et al details their development of a microbioreactor system with integrated oxygen measurements via optodes. The authors describe their system (with appropriate supporting information to enable fabrication) and the extensive calibration and determination of system parameters. Important for a paper of this type, the methods section is very clearly written with a suitable amount of detail - one of the best written methods section I've reviewed recently. Overall, all of the experiments the authors completed to analyze their system are appropriate, and the overall presentation of the work is excellent. While I have some minor concerns and suggestions (detailed below), the manuscript is certainly suitable for publication with some minor changes (the most important being incorporation of statistical tests to assay significance of differences) and I look forward to it's publication.

We thank the reviewer for their in-depth review and constructive criticism.

Minor Issues & Suggestions

- Several sections would benefit from statistical analysis. Basically anywhere where different conditions are being compared (e.g. fig 3, fig 4, fig 5c, possibly fig 6, fig 7, fig 8, some SI figures, and accompanying test. Just doing a comparison and adding if it is significant or not would help to clarify. E.g. - p19 - "reports approximately 4% lower values" but I would guess not significantly different.

This is a great suggestion. We used a Kolmogorov-Smirnov test to determine if any pair of calibrations (“base” and “treatment”) were significantly different. We have added a few lines to the end of the methods section describing the statistical method used, and added the results of those tests to sections 3.1.1, 3.1.2, 3.1.3, and 3.1.6. In all cases (as the reviewer expected) the statistics backed up the existing conclusions.

- Probably worth changing "fluorescent dye" etc. to luminescent given the mechanism for the dyes

We appreciate this suggestion; however the commercial optical oxygen sensors are predominantly referred to as “fluorescent”. As such, we prefer to stick with that term to hopefully make it easier for domain scientists to find our work.

- Possibly worth cutting 2.5.7 and 2.5.8 from the methods section and integrating into the results and discussion. Not much actual "methods" and more explanation as for motivation.

We have moved the “motivation” aspects of 2.5.7 to results and discussion section 3.1.7. We have left 2.5.8 as it was, since it does describe the process for creating direct-coated vials, though not in a great deal of detail.

- Figure 5 - possibly add labels to figure (e/g. "autoclaved""not autoclaved" to image) to help make it easier to understand at a glance

Labels differentiating the “autoclaved” and “not autoclaved” optodes have been added to the figure.

- P17 (section 3.1.4) "shows that the oxygen permeability into the reactors is low" - add range for what "low" is or explain to the readers. I.e. maybe worth comparing with normal units and show that it is considered low?

Rather than provide normal units for permeability and compare to specific materials (we feel that this would not provide much actionable information), we have added text to explain that we mean “low” in the context of the oxygen consumption rates that are likely to be measured with this system, and some more discussion about when oxygen permeability is likely to be a concern: “In general, Table 2 shows that oxygen permeability into the reactors is low relative to many of the oxygen consumption rates reported in environmental studies, and in many experimental setups dealing primarily with higher oxygen concentrations, faster consumption rates, and/or shorter experiment times it may be safe to ignore entirely. This is less likely to be the case in systems with very low oxygen concentrations and/or very low oxygen consumption rates.”

- Figure 6 (and other may benefit too) - smaller data points to show individual points more clearly.

The data points in figure 6 have been made smaller. 

- Section 3.1.5 mentions click chemistry detailed in methods, but I didn't see it there (may have just missed it).

This was not explained clearly. The click-chemistry process is only provided via reference in the methods section, not detailed in this manuscript. To make this clearer, we have revised the sentence in 3.1.5 and replicated the pertinent reference there. “performance is likely available either by using the Click-chemistry dye process referred to in the methods section”

- Probably not worth it, but maybe comparing with a different camera type in addition to a duplicate camera would add value for the reader (i.e. same camera - calibration probably fine, different camera - new calibration needed).

This is a great suggestion; however it is beyond the scope of this manuscript. We have added a line in the Summary, Limitations, and Next Steps section to indicate that evaluation variation between different camera manufacturers and models would be useful.

---

## [Decision Letter · Decision Letter 1]

22 Jun 2023

Metabolic Multireactor: practical considerations for using simple oxygen sensing optodes for high-throughput batch reactor metabolism experiments

PONE-D-23-06927R1

Dear Dr. Matthew Kaufman,

We’re pleased to inform you that your manuscript has been judged scientifically suitable for publication and will be formally accepted for publication once it meets all outstanding technical requirements.

Kind regards,

Catarina Leite Amorim, Ph.D.

Academic Editor

PLOS ONE

Additional Editor Comments (optional):

All comments have been carefully addressed by authors, improving the manuscript quality and significance.

Reviewers' comments:

Reviewer's Responses to Questions

**Comments to the Author**

Reviewer #1: All comments have been addressed

Reviewer #2: All comments have been addressed

2. Is the manuscript technically sound, and do the data support the conclusions?

Reviewer #1: Yes

Reviewer #2: Yes

3. Has the statistical analysis been performed appropriately and rigorously? 

Reviewer #1: Yes

Reviewer #2: Yes

4. Have the authors made all data underlying the findings in their manuscript fully available?

Reviewer #1: Yes

Reviewer #2: Yes

5. Is the manuscript presented in an intelligible fashion and written in standard English?

Reviewer #1: Yes

Reviewer #2: Yes

6. Review Comments to the Author

Reviewer #1: The authors have adressed all my comments. I would like to congratulate the authors once more for this nice method and I am looking forward to see future outcomes.

Reviewer #2: (No Response)

7. PLOS authors have the option to publish the peer review history of their article (what does this mean?). If published, this will include your full peer review and any attached files.

Reviewer #1: No

Reviewer #2: No

---

## [Editor Report · Acceptance letter]

3 Jul 2023

PONE-D-23-06927R1 

Metabolic Multireactor: practical considerations for using simple oxygen sensing optodes for high-throughput batch reactor metabolism experiments 

Dear Dr. Kaufman:

I'm pleased to inform you that your manuscript has been deemed suitable for publication in PLOS ONE. Congratulations! Your manuscript is now with our production department. 

Kind regards, 

on behalf of

Dr. Catarina Leite Amorim 

Academic Editor

PLOS ONE